# Proposal for Structured Histopathology of Nasal Secretions for Endotyping Chronic Rhinosinusitis: An Exploratory Study

Stephan Vlaminck [1],*, Emmanuel Prokopakis [2], Hideyuki Kawauchi [3], Marc Haspeslagh [4], Jacques Van Huysse [5], João Simões [1], Frederic Acke [6] and Philippe Gevaert [6]

1 Department of Otorhinolaryngology, Centre Hospitalier de Mouscron, B-7700 Mouscron, Belgium
2 Department of Otorhinolaryngology, School of Medicine, University of Crete, G-14122 Crete, Greece
3 Department of Otorhinolaryngology, University Hospital, Shimane, Matsue 693-8501, Japan
4 Dermatology Research Unit, Ghent University Hospital, B-9000 Ghent, Belgium
5 Department of Pathology, AZ St-Johns Hospital, B-8000 Bruges, Belgium
6 Department of Otorhinolaryngology, Ghent University Hospital, B-9000 Ghent, Belgium
* Correspondence: stephan.vlaminck@gmail.com

**Abstract:** Background: The EPOS guidelines promote cellular analysis as a primary goal in endotyping chronic rhinosinusitis (CRS). Current analysis is mainly based on biopsy or operative tissue collection, whereas the use of sinonasal secretions for inflammatory endotyping is not advocated in clinical practice. Early endotyping is crucial though, especially regarding the increasing evidence of patient-tailored therapy. We aimed to investigate the diagnostic value and reproducibility of sinonasal secretions sampling. Methods: First, preoperative secretion analysis of 53 Caucasian CRS patients was compared to subsequent operative tissue analysis. Second, secretion analysis at two different time points was compared for 10 postoperative Caucasian CRS patients with type 2 (T2) inflammation and 10 control participants. Secretions were collected by both endoscopic aspiration and nasal blown secretions in all participants. Results: The sensitivity to detect T2 inflammation was higher in nasal aspiration samples (85%) compared to nasal blow secretions (32%). A specificity of 100% for both techniques was obtained. A 90% reproducibility for T2 eosinophil detection was found by sampling at different time points regardless of the technique. Of the T2 patients, 60% showed no T2 inflammatory pattern more than one year after endoscopic sinus surgery. Conclusions: Nasal secretion sampling, especially aspiration of nasal secretions, is useful in the detection of T2 inflammation in CRS pathology. We proposed a structured histopathology analysis to be useful in daily clinical practice, which includes Congo red staining sensitive for eosinophilic cells and free eosinophil granules. Analysis of nasal secretions enables endotyping in an early stage, allowing more directed therapy.

**Keywords:** chronic rhinosinusitis; nasal polyps; CRSwNP; T2 inflammation; endotyping; nasal secretions; endoscopic sinus surgery; histopathology; tissue analysis

## 1. Introduction

Chronic rhinosinusitis (CRS) is characterized by persistent symptomatic inflammation of the nasal and paranasal mucosa lasting longer than 12 weeks [1]. CRS is generally divided into two major clinical phenotypes: chronic rhinosinusitis with and without nasal polyps (CRSwNP and CRSsNP respectively). The wide range of inflammatory patterns together with mucociliary and/or structural abnormalities has resulted in an attempt to define endotypes based on cytologic histopathology.

The EPOS guidelines sustain the mucosal concept as a primary diagnostic step defining different inflammatory clusters as endotypes, such as T helper 1 (T1)-driven or neutrophilic inflammation, and T helper 2 (T2)-driven or eosinophilic inflammation [2], including both innate and adaptive immunity. In their 2020 report, the EPOS steering group concluded that the amount of eosinophilic infiltration and the overall intensity of the inflammatory

response were closely related to the prognosis and severity of disease and their response to biologicals [1], implicating the need for cellular analysis. They advised microscopic analysis of surgically obtained tissue, whereas secretions analysis at initial presentation was not considered an essential measure in the CRS work-up, despite the impact of the inflammatory type on prognosis and therapy. Moreover, a recommendation about the need for cytology after failure of medical or surgical treatment despite persistent symptoms and abnormal mucosa at endoscopy was not given [1]. In the JESREC study, Tokunaga et al. also found a strong correlation between prognosis after surgery and mucosal eosinophilia obtained during surgery [3]. The latter was not included in the JESREC criteria though because a prognosis estimation is mainly relevant before eventual therapy.

The timing and methodology of assessing the endotype is not unequivocally defined in clinical guidelines. Snidvongs et al. proposed structured histopathology as the method of CRS profiling in routine practice, thereby emphasizing the importance of the presence of eosinophils and eosinophil aggregates [4]. The latter have been described in nasal secretions (mucin) and was associated with significantly worse endoscopic scores [4]. Meanwhile more studies used this histopathological concept to show the presence of T2 endotype inflammation by eosinophil aggregates [5–7]. T2 inflammation confirmed by eosinophil mucin could be associated with worse surgical outcomes and recurrent or persistent disease at long-term follow-up [8,9], and even predicted the onset or aggravation of T2-inflammatory asthma [10]. Consequently, the above-mentioned studies suggested that analysis of secretions for eosinophils and eosinophilic aggregates can be performed in daily practice and even prior to surgery.

Cell counting in nasal secretions has been described in nasal lavage, mucosal scrapings, and endonasal biopsies. These are considered more invasive procedures. In contrast, Nair et al. demonstrated the usefulness of eosinophil count from nasal blown secretions in the treatment of atopic patients [11]. Sputum eosinophil counts have also been proven a reliable tool in controlling asthma exacerbations [12]. Koenderman et al. discussed the lack of comparable results between eosinophil numbers in blood and sputum versus tissue concentrations. They also questioned the use of blood analysis as a diagnostic marker and/or monitoring tool for biological responses in eosinophilic asthma treatment [13]. Ueki et al. stated that the circulating eosinophil count does not always reflect tissue eosinophilia and vice versa, and doubted the validity of results between two different compartments [14].

The handling of sinonasal secretions samplings for laboratory-based studies has been summarized before [15,16]. Nasal secretions can either be collected in bulk by nasal blown secretions (NBS); endoscopic aspiration of nasal secretions (ANS), nasal washings; or by collecting with filter paper, cotton wool, foam, or sponges. The relatively low cost and non-invasive sampling procedure with subsequent structured histopathology of NBS and ANS functioned as the origin of this exploratory study. We aimed to evaluate the validity, feasibility, and reproducibility of cell counting as a diagnostic procedure and the value of its routine clinical evaluation, by performing NBS and ANS in CRS patients. As a secondary outcome, we questioned whether the secretion sampling results of patients who were operated on for T2 CRS more than one year before inclusion, still showed T2 characteristics using the two collecting techniques.

## 2. Materials and Methods

### 2.1. Patient Inclusion

For the first experiment (validity), adult patients with the indication of endoscopic sinus surgery because of CRSsNP or CRSwNP were invited to participate. Patients were included between 07/2021 and 12/2021 and patient characteristics were collected (age, gender, CRS phenotype, allergy, and asthma). Patients were asked to undergo endoscopic aspiration of nasal secretions by the ENT surgeon and to collect a nasal blown secretion sample themselves. The type of inflammation (T1 or T2) and the number of eosinophils per high power field (HPF) were subsequently determined on microscopic analysis of sinonasal

tissue obtained during surgery [1]. In the time frame between nasal secretions sampling and endoscopic sinus surgery, no drug treatment alterations were implemented, nor were any oral steroids used or added.

For the second experiment (reproducibility), adult patients with T2 eosinophilic fungal rhinosinusitis (EFRS) pathology were compared to control patients. The pathology group underwent endoscopic sinus surgery minimal one year before inclusion in the study and proved T2 EFRS based on operative tissue analysis. They had local clinical and endoscopic control of disease with saline irrigation and/or local steroids. The control patients were patients without acute or chronic airway inflammatory disease clinically and endoscopically. The only rhinological symptoms they had experienced in the past were temporary symptoms matching a common cold. All participants were otherwise healthy. The experiment was performed in the year 2021. All participants collected a nasal blown secretion sample themselves in the morning and underwent endoscopic aspiration of nasal secretions by the surgeon on the same day in the evening. These measures were repeated after 3–4 days, for a total of four samplings per participant.

The study was performed in Belgium, Europe. Both experiments were approved by the Ethics Committee AZ St Jan Hospital (Bruges, Belgium; internal number 2349). All patients participated after informed consent.

*2.2. Sinonasal Sampling*

The technique of collecting nasal blown secretions was demonstrated first and subsequently performed by the patient at home. After waking up, the patients were asked to blow their nose in a recipient (e.g., a coffee cup). A ThinPrep CytoLyt solution of 30 mL (Hologic Inc., Marlborough, MA, USA) was added to the nasal blown secretions until completely embedded. This was preferred over the toxic buffered formaldehyde. The mixture was transferred to the Cytolyt vial and returned to the lab within 24 h.

Endoscopic aspiration of sinonasal secretions in the middle meatus was performed by one ENT surgeon (39 y of ENT experience) in a consultation setting using a bronchopulmonary aspiration collector device (ref. 24001182, ConvaTec, Reading, UK), after which a Cytolyt vial of 30 mL was added. The sample was brought to the lab for analysis within 24 h.

*2.3. Laboratory Analysis*

After embedding in paraffin, the sample was cut into 4 μm sections and stained with haematoxylin and eosin (H&E). Eosinophil cell counts were considered positive when 10 or more eosinophils were seen per HPF ($\times 400$). The number of eosinophils of the positive patients were divided into the following categories: 10–49, 50–99 and >99.

For the second experiment, all specimens were further studied for the presence of layers of eosinophils, free eosinophil granules (FEGs), and Charcot Leyden crystals (CLC, debris of dead eosinophils). An adjunctive Congo red colouring was performed in all participants to better identify eosinophils, as well as an adjunctive Gomori methenamine silver staining (GMS) of the specimens, in search of fungal hyphae (FH). All laboratory analyses were performed by one pathologist (37 y of pathology experience) who was blinded for the participants' phenotypes, and one experienced researcher. Disagreements were resolved by consensus after discussion.

**3. Results**

*3.1. Validity of Secretions Sampling*

The first experiment comprised the inclusion of 53 CRS patients who were scheduled to undergo endoscopic sinus surgery for the first time and were preoperatively evaluated by nasal blown secretions and nasal aspiration. Of the included patients, 12 had a T1 type of inflammation and 41 a T2 type. Their baseline characteristics can be observed in Table 1, divided into the T1 or T2 endotype based on tissue analysis. Notably, all patients with a T1 inflammation had CRSsNP, and all but one with T2 inflammation had CRSwNP.

**Table 1.** Characteristics of patients included in the first experiment.

|  | **T1 Inflammation** | **T2 Inflammation** |
|---|---|---|
| Number of patients | 12 | 41 |
| Age (mean, range) | 51.6 y (18–66 y) | 55.6 y (33–86 y) |
| Gender | 4 F–8 M | 15 F–26 M |
| Phenotype | 12 CRSsNP | 1 CRPsNP–40 CRSwNP |
| Allergy | 2/12 (16.7%) | 21/41 (51.2%) |
| Asthma | 1/12 (8.3%) | 26/41 (63.4%) |

All patients with a T1 inflammation had a negative eosinophil count (<10 eosinophils/HPF) on all three analysis methods: operative tissue collection, preoperative nasal blown secretions, and preoperative nasal aspiration. The eosinophil count for the patients with a T2 inflammation is provided in Table 2. When the results of nasal blown secretions were compared with nasal aspiration, fewer positive eosinophil counts were observed in the former. All positive nasal blown secretions samples were positive on sinonasal aspiration as well. Sensitivity and specificity for nasal blown secretions was 31.7% and 100%, respectively; for nasal aspiration 85.4% and 100% were found, respectively. The positive predictive value both techniques was 100%.

**Table 2.** Eosinophil count of patients with T2 inflammation included in the first experiment for the three different sampling methods.

|  | **Preoperative Nasal Blown Secretions** | **Preoperative Aspiration of Nasal Aspiration** | **Surgical Tissue** |
|---|---|---|---|
| <10 eos/HPF | 28 (68.3%) | 6 (14.6%) | 0 (0.0%) |
| 10–49 eos/HPF | 6 (14.6%) | 15 (36.6%) | 5 (12.2%) |
| 50–99 eos/HPF | 6 (14.6%) | 13 (31.7%) | 19 (46.3%) |
| >99 eos/HPF | 1 (2.4%) | 7 (7.1%) | 17 (41.5%) |

### 3.2. Reproducibility of Secretions Sampling

The second experiment comprised 10 proven T2 EFRS patients and 10 controls. The mean age of the patients and controls was 56.9 y and 38.3 y, respectively (range 42–79 y and 27–69 y, respectively. There were six female and four male patients, and four female and six male controls. Table 3 shows the presence of eosinophils and neutrophils for the two groups on NBS and ANS for the two different time points (3–4 days apart). Of the T2 EFRS patient group, 60% showed no T2 inflammatory pattern more than one year after endoscopic sinus surgery, whereas T2 inflammation seemed to persist in 40% regardless of the sampling technique.

**Table 3.** Eosinophil and neutrophil positivity of T2 EFRS patients and controls, based on nasal blown secretions (NBS) and aspiration of nasal secretions (ANS) at two different time points.

|  |  | **NBS Day 0** | **NBS Day 3–4** | **ANS Day 0** | **ANS Day 3–4** |
|---|---|---|---|---|---|
| T2 EFRS patients | Eosinophils | 4 (40%) | 4 (40%) | 4 (40%) | 4 (40%) |
|  | Neutrophils | 3 (30%) | 4 (40%) | 4 (40%) | 4 (40%) |
| Controls | Eosinophils | 2 (20%) | 0 (0%) | 3 (30%) | 1 (10%) |
|  | Neutrophils | 7 (70%) | 8 (80%) | 4 (40%) | 4 (40%) |

When the two different time points were compared, the same result was obtained in 18/20 (90%) for eosinophil and 13/20 (65%) for neutrophil presence in NBS. When considering ANS, we found the same result in 18/20 (90%) for eosinophil and 18/20 (90%) for neutrophil presence.

### 3.3. Laboratory Staining Techniques

### 3.3.1. Haematoxylin and Eosin Staining (H&E)

Of the four eosinophil-positive EFRS patients, both NBS and ANS samplings showed the presence of >99 eosinophils per HPF, the presence of mucin in which necrotic eosinophils, FEGs, and CLC were observed. In some of these, even FH could be observed by H&E staining. Mucin or FH was not present in the eosinophil-positive controls, but a combination with neutrophil cells was possible (Figure 1).

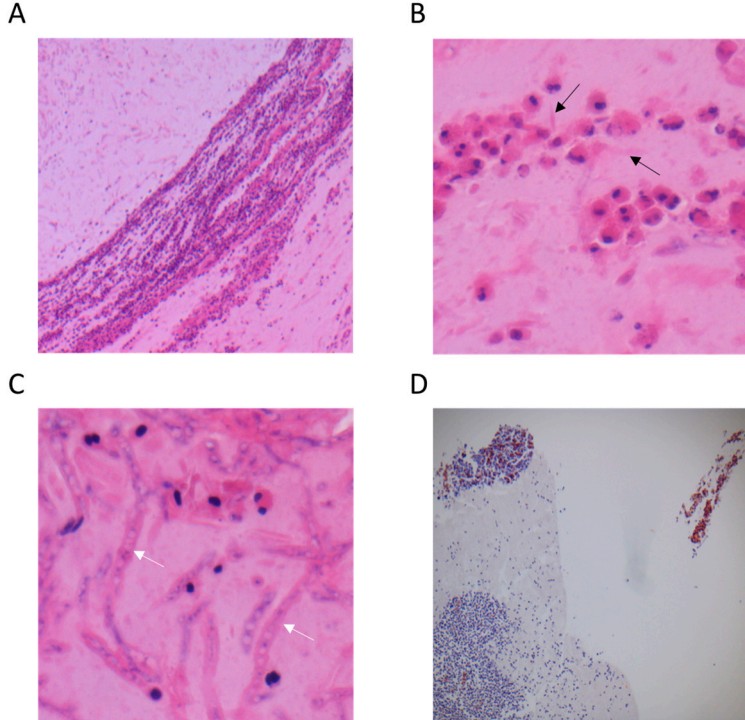

**Figure 1.** Images from H&E staining of samples obtained by the aspiration of nasal secretions technique. (**A**) A layered aspect of eosinophils aggregated in mucin, and abundant granules (50× magnification); (**B**) CLC (black arrows) surrounded by granules and necrotic eosinophil cells (400× magnification); (**C**) Presence of fungal hyphae (400× magnification); (**D**) Combination of patches of eosinophil and neutrophil cells (10× magnification).

### 3.3.2. Congo Red Staining

Abundant presence of eosinophilic cells (>99 eosinophils per HPF) were found in the four eosinophil-positive EFRS patients by Congo red staining. The presence of red colouring FEGs in all these samplings confirmed their proteinic content (Figure 2). Of interest, one eosinophil-positive control also showed eosinophils and FEGs on Congo red staining, whereas the others did not.

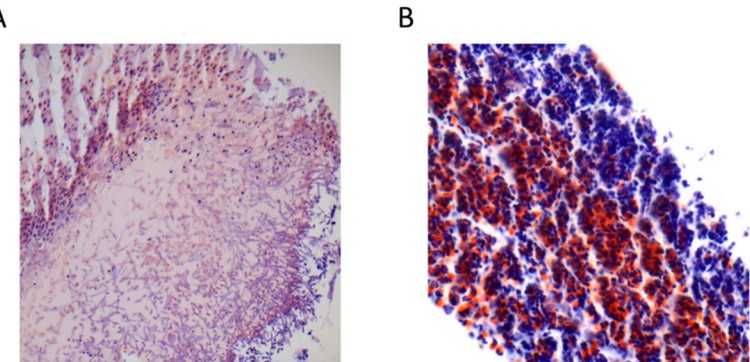

**Figure 2.** Images from Congo red staining. (**A**) Eosinophils, FEGs and FH (sample obtained by the aspiration of nasal secretions technique, 100× magnification); (**B**) Eosinophils, focal neutrophils, and FEGs (sample obtained by the nasal blown secretions technique, 200× magnification).

### 3.3.3. Gomori Methenamine Silver Staining (GMS)

Only two of the four eosinophil-positive EFRS patients showed the presence of fungal hyphae on GMS staining (Figure 3), compared to none of the eosinophil-negative EFRS patients and none of the controls.

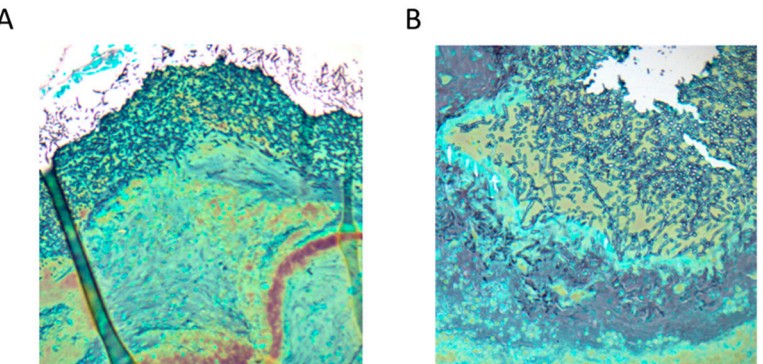

**Figure 3.** Images from Gomori methenamine silver staining (GMS) of samples obtained by the aspiration of nasal secretions technique. (**A**) Massive presence of fungal hyphae (50× magnification); (**B**) Massive presence of fungal hyphae (100× magnification).

## 4. Discussion

With this exploratory study, we aimed to evaluate the validity, feasibility, and reproducibility of nasal secretions sampling for structured histopathology, in the endotyping of CRS patients. First, we found that a T2 inflammatory endotype detected by aspiration of sinonasal secretions or nasal blown secretions was associated with tissue T2 inflammation in all cases. The sensitivity to detect T2 inflammation was higher in nasal aspiration samples (85%) compared to nasal blow secretions (32%). Second, 90% reproducibility for T2 eosinophil detection was found by sampling at different time points regardless of the technique. Finally, we showed that 60% of the T2 EFRS patients showed no T2 inflammatory pattern more than one year after endoscopic sinus surgery, whereas T2 inflammation seemed to persist in 40% regardless of the sampling technique.

In the first experiment, the specificity and sensitivity of ANS and NBS for T2 detection was calculated, in order to assess its value compared to the more invasive tissue analysis. We found a specificity of 100% and a sensitivity of 85% for ANS, and a specificity of 100% and sensitivity of 32% for NBS. Consequently, if T2 inflammation is found by nasal secretion sampling, it is assumed that these patients do have local T2 inflammation, which might imply specific counselling and therapeutical options. However, if T2 inflammation is not found, we cannot completely exclude T2 involvement. Moreover, a suction device-applied sampling performed by the ENT surgeon seems more efficient than the patient-dependent

nasal blown secretions. Possible explanations might be patient-related factors and mucin adherence might be better overcome by nasal aspiration. All patients underwent both techniques without any side effects. NBS was previously assessed as a technique with reproducible and reliable results in allergic patients [11]. By using CytoLyt, we observed that the sinonasal secretions remained intact over at least 24 h, maintaining the observed cohesion, which is important in the detection of DNA-mediated cluster aggregates. The presence of those aggregates of clustered cells can be explained by the viscosity based on the presence of chromatin [17].

In the second experiment, we found a 90% reproducibility for T2 eosinophil detection by sampling at different time points regardless of the technique. However, the same was done for neutrophil detection, with a 65% reproducibility for NBS, but 90% for ANS. This can be explained by the time of sampling and by the variability of neutrophilic expression. Neutrophilic presence is easily influenced by environmental factors and might explain the variability of neutrophil counting over time. The higher neutrophilic variability of NBS might also be explained by the time of sampling: NBS was collected in the morning and ANS rather towards the evening. In the morning, secretions reflect undisturbed production over hours, whereas in the evening, secretions are probably more recently produced.

In the same experiment, we observed that 60% of the T2 EFRS patients, as assessed during endoscopic sinus surgery for CRSwNP, showed no T2 inflammatory pattern on secretions at least one year after surgery. Consequently, by adequate treatment, T2 EFRS patients who are prone to aggressive recurrence remained in a T2 remission state for at least one year. It was recently shown that CRSwNP patients with normal cytology by nasal scraping at follow-up evaluation, had a lower probability to have a first recurrence episode within 10 years (59% of patients) [18]. The remaining patients showed a higher CRSwNP recurrence risk. For our patient group, the 40% of patients with eosinophil-positive nasal secretions thus remain at higher risk of developing nasal polyp recurrence and could be followed up more closely.

A structured histopathology of local tissue was proposed by Snidvongs et al. and Brescia et al., amongst others [4,19]. In addition, the former emphasized the importance of nasal secretion analysis, especially for the presence of CLC, FH, and eosinophil aggregates [4]. Aggregates have been described in tissue, but can also be observed in nasal secretions, where they are often named eosinophilic mucin. Eosinophilic mucin is an aggregate of intact eosinophils and of eosinophils that underwent a non-apoptotic cell death pathway, namely extracellular trap cell death (ETosis) that mediates an active eosinophil cytolytic degranulation [17]. In this way, free eosinophil granules (FEGs) can be observed in eosinophilic mucin, hence underlining a T2-drive eosinophilic inflammation [20,21]. The presence and role of free granules was already described by Persson et al., suggesting an active inflammation pattern in vivo and possibly a driving force in the inflammatory cascade [22]. The same group also suggested a possible effect of anti-IL-5 therapy in patients with extensive degranulation [23] and stressed the importance of FEGs in asthma [24].

Granule proteins can be quantified, including eosinophil cationic protein, major basic protein, and eosinophil peroxidase. These proteins typically colour red under microscope with Congo red staining. The presence of a major basic protein is a marker for T2 CRS and seems to be totally lacking in the T2-driven allergic pathway disease [25]. Moreover, major basic protein production has been shown to be related only to the eosinophilic cells [26]. To the best of our knowledge, we are the first to include a Congo red staining in order to visualize intact eosinophils and FEGs in nasal secretions in all four eosinophil-positive T2 EFRS patients.

CLC can be observed by a H&E staining and was also shown in all four eosinophil-positive T2 EFRS patients. Nowadays, the identification of CLC is a reliable parameter for T2 inflammatory presence [27,28]. Notably, eosinophils, FEGs and CLC were also found in one young control participant without symptoms or endoscopic pathology to date. We might hypothesize a subclinical stage of T2 inflammatory pathology in this participant.

Another observation was the combined presence of T1 and T2 cellular aggregates in a patchy appearance where conglomerates of neutrophilic cells can be seen together with T2 aggregates (Figure 1D). This corresponds with the findings of Delemarre et al., who observed neutrophil inflammation as a regular part of severe T2 CRS [29]. Congo red staining will not occur when neutrophilic inflammation is present except in these combination cases. Adjuvant Congo red staining therefore may be considered a useful diagnostic tool to distinguish T2 eosinophilic inflammation, including FEGs, with a T1 driven mechanism. The proposal for structured histopathology of nasal secretions for endotyping CRS as performed in the current study is provided in Figure 4.

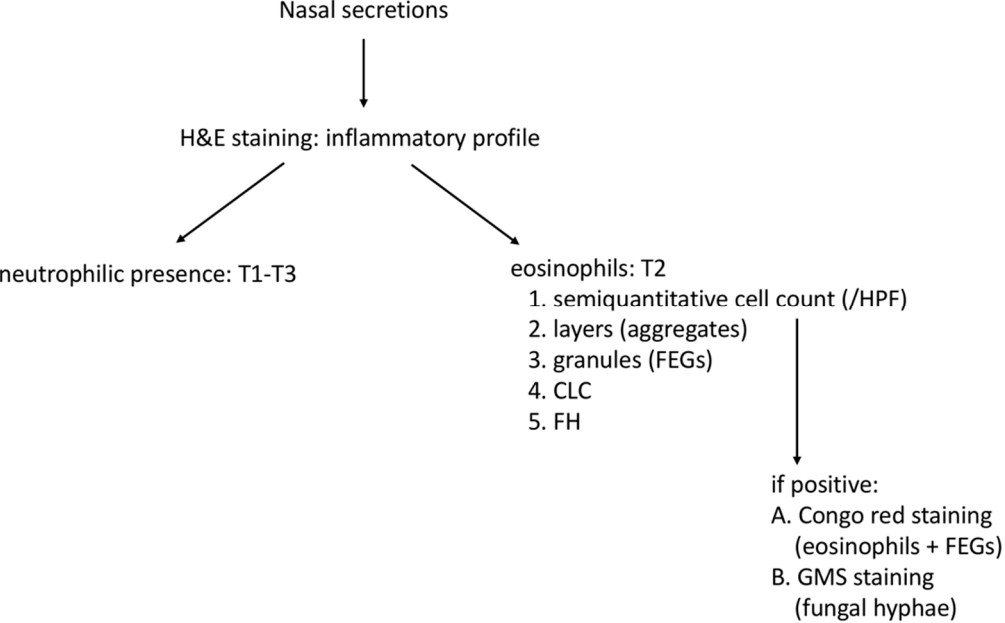

**Figure 4.** Proposal for structured histopathology of nasal secretions for endotyping CRS.

Although the EPOS 2020 guidelines claimed the importance of endotyping in CRS patients [1], endotyping by tissue analysis, often during surgery, comes late in the diagnostic work-up. Obtaining secretions in a consultation setting might be a non-invasive and low-cost way to assess the type of inflammation early, allowing the ENT clinician to obtain valuable information relevant for further steps of treatment. Endotype-driven pathways are being developed with biologicals, amongst others. Moreover, the surgical efficiency of CRSwNP seems lower when performed later in the disease [30], suggesting an early decision might improve outcomes. In CRSsNP, recurrence after endoscopic sinus surgery was shown in 9% of patients during a minimal follow-up of three years, compared to 40% in CRSwNP [9]. Moreover, 48% recurrence was found in eosinophil-positive CRSwNP, elevating to 73% in the presence of eosinophils, eosinophilic mucin, and fungal hyphae [9]. In patients with a high risk of recurrence, such as T2, more elaborate surgery, aimed at large and open sinuses for improved local therapy postoperatively, should be considered [31].

As this is an exploratory study, our sample sizes are rather low, but suggest that nasal secretions are an important tool in the early endotyping of CRS and implementation on a larger scale. Measuring specific cytokine levels in nasal secretions could strengthen the T1/T2 status, but will also result in an additional cost. In contrast with the first experiment, the postoperative nasal secretion sampling results of the second experiment were not compared with nasal tissue biopsies obtained at the same time, as these patients did not need surgery. Finally, we should be aware of racial differences in CRS endotyping: CRSwNP in Caucasians is more T2-driven than in Asians [1].

## 5. Conclusions

In conclusion, this exploratory study shows that nasal secretion sampling is useful in the detection of T2 inflammation in CRS pathology. Aspiration of nasal secretions by the ENT surgeon demonstrated a more beneficial profile versus nasal blown secretions. We proposed a structured histopathology analysis applied to sinonasal secretions useful in daily clinical practice, including a Congo red staining technique sensitive to show eosinophilic cells and free eosinophil granules. Nasal secretion sampling enables endotyping in an earlier stage than the proposed diagnostic scheme of the EPOS 2020 guidelines. In terms of precision medicine, early endotyping might allow more directed medical therapy and a patient-tailored surgical approach, if needed.

**Author Contributions:** Conceptualization, S.V., E.P., H.K. and P.G.; methodology, S.V., F.A. and P.G.; formal analysis, S.V. and F.A.; clinical investigation, S.V. and J.S.; histopathologic analysis, S.V., M.H. and J.V.H.; writing—original draft preparation, S.V. and F.A.; writing—review and editing, S.V., H.K., M.H., F.A. and P.G.; visualization, S.V. and M.H.; supervision, E.P., H.K. and P.G. All authors have read and agreed to the published version of the manuscript.

**Funding:** This research received no external funding.

**Institutional Review Board Statement:** The experiments were approved by the Ethics Committee AZ St-Johns Hospital (Bruges, Belgium). The study was conducted in accordance with the Declaration of Helsinki.

**Informed Consent Statement:** Informed consent was obtained from all subjects involved in the study.

**Data Availability Statement:** The raw data can be provided on simple request.

**Acknowledgments:** We would like to thank all participants.

**Conflicts of Interest:** The authors declare no conflict of interest. E.P. and H.K. are editors of Allergies' special issue 'Recent advances in allergic rhinitis'.

## Abbreviations

ANS: aspiration of nasal secretions; CLC: Charcot Leyden crystals; CRS: chronic rhinosinusitis; CRSsNP: CRS without nasal polyps; CRSwNP: CRS with nasal polyps; EFRS: eosinophilic fungal rhinosinusitis; FEGs: free eosinophil granules; FH: fungal hyphae; GMS: Gomori methenamine silver staining; H&E: haematoxylin and eosin; HPF: high power field; NBS: nasal blown secretions; T1: type 1 inflammation; T2: type 2 inflammation.

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
