# Peer review of "Proposal for Structured Histopathology of Nasal Secretions for Endotyping Chronic Rhinosinusitis: An Exploratory Study"

_allergies, doi:10.3390/allergies2040012_

Round 1

Reviewer 1 Report

This is a an exploratory study about proposal for structured histopathology of nasal secretions for 2 endotyping chronic rhinosinusitis.The authors study the  value of structured histopathology of nasal secretions for chronic rhinosinusitis. This is an interesting paper which addressed an important area of chronic rhinosinusitis research. However, there are something that needed to clarify before drawing some conclusions. 

1.As a clinical research, in addition to ethics, it is best to have clinical registration.

2.There are many versions of eosinophil count standards for eosinophilic sinusitis. Which version did the researchers choose? And list the relevant reasons.

3.There is too much content in the introduction part, only need to explain the relevant content.

4.In immunohistochemical images, it is not enough to have positive images, but need to have control group or negative images. Please also ensure that a scale bar is included with every microscopy image, and magnification in the figure legends.

Author Response

Dear Reviewer

Thank you for your critical reading and suggestions to improve the paper. You may find the answers to your questions below.

Question 1: As a clinical research, in addition to ethics, it is best to have clinical registration.

Answer: We agree that registration of our clinical study in a databank would have been useful at initiation and will do this in future studies. 

Question 2: There are many versions of eosinophil count standards for eosinophilic sinusitis. Which version did the researchers choose? And list the relevant reasons.

Answer: As the study is conducted in Caucasians, we followed the EPOS 2020 guidelines defining a positive T2 cellular endotype by 10 eosinophils or more per high power field (400x magnification) based on histopathology with semiquantitative counting. The reference has been added to the Methods section. We are aware of other standards (e.g. cut-off of 70 eosinophils/HPF in the JESREC study aiming at the selection of severe T2 inflammation related to prognosis) but felt that the EPOS standards met our study best (intention, race...).

Question 3: There is too much content in the introduction part, only need to explain the relevant content.

Answer: We have streamlined the introduction part, thereby reducing the word count from 880 words to 660 words.

Question 4: In immunohistochemical images, it is not enough to have positive images, but need to have control group or negative images. Please also ensure that a scale bar is included with every microscopy image, and magnification in the figure legends.

Answer: We did not provide negative images as they imply absence of significant cellular content, or much lower than 10 cells per high power field. These images do not seem appropriate to include in the published manuscript but can be provided upon request; we hope you will agree with this. The magnification of each figure has been added to the figure legends.

Reviewer 2 Report

The manuscript by Vlaminck et al describes the use of structured histopathology of nasal secretions to endotype patient with chronic rhinosinusitis in order to assist with patient selection and monitoring in the use of new biological therapies in CRSwNP. The idea is interesting and novel. Please find my detailed comments below: 

General Comments:

The literature review in the introduction has been done very thoroughly however in my opinion it is overly long and therefore it would benefit from streamlining.

I am not quite clear on how T2 and T1 status was defined. From the methodology I gather it was based on Eosinophil counts in the blown nose and aspiration samples. Is this correct? Was it only based on Eosinophil numbers? Were all patients defined to be T2 later confirmed with the surgical biopsies? How did your definition of T2 status correlate with peripheral blood measurements of Eosinophil numbers and ECP levels? If T2 status was purely defined by eosinophilia as the authors saw Eosinophils in the control patients without disease I do not believe can claim that this is a reliable technique to diagnose T2 status.

Would it be possible for the authors to measure T1 and T2 cytokine levels in the nasal secretions? This in conjunction with the eosinophil counts would help to strengthen their claims of T2 or T1 status. If not this limitation should be discussed in the discussion.

For Acronyms that are repeated less than 10 times please consider writing them out in full. It is quite tiring for the reader to have to look up acronyms all the time. Or if preferred the authors could provide a list a used acronyms throughout the manuscript.

Regarding Figures 1,2 and 3 from what source are these microscopy images from? Are they representative from ANS or NBS? Please include this in the figure legends

Specfic Comments:

Line 129: „No drug treatment changes were applied from 129 secretions sampling to ESS.“ This sentence is not very clear. Do you mean no disease specific drug treatments were initiated between nasal secretion sampling and surgery?

Line 135: „The control patients were patients without chronic airway inflammatory disease clinically and endoscopically, but eventually a common cold“ does not make sense and this is the wrong use of eventually. I presume the authors mean some of the control group had suffered symptoms of a common cold in the past year?

Please define what is meant by the acronym EFRS.

Throughout the article the authors repeat „on two different time points“ the correct preposition to use here is „at“ not „on“

Line 206: please state in the text when these timepoints were. From table 3 I gather the samples were taken 3-4 days apart. Is this correct?

Author Response

Dear Reviewer

Thank you for your critical reading and suggestions to improve the paper. You may find the answers to your questions below.

Question 1: The literature review in the introduction has been done very thoroughly however in my opinion it is overly long and therefore it would benefit from streamlining. 

Answer: We have streamlined the introduction part, thereby reducing the word count from 880 words to 660 words.

Question 2: I am not quite clear on how T2 and T1 status was defined. From the methodology I gather it was based on Eosinophil counts in the blown nose and aspiration samples. Is this correct? Was it only based on Eosinophil numbers? Were all patients defined to be T2 later confirmed with the surgical biopsies? How did your definition of T2 status correlate with peripheral blood measurements of Eosinophil numbers and ECP levels? If T2 status was purely defined by eosinophilia as the authors saw Eosinophils in the control patients without disease I do not believe can claim that this is a reliable technique to diagnose T2 status.

Answer: The T1 or T2 diagnosis was only made based on surgical biopsies. In the first experiment, cell counting of preoperative nasal blown secretions and aspiration of nasal secretions was indeed compared to those of surgical biopsies (the latter considered the gold standard). All samples were analyzed based on semiquantitative counting, with a T2 diagnosis in presence of ten or more eosinophils per high power field (magnification x400) on the surgical biopsy sample, as defined in the EPOS 2020 guidelines. In the second experiment, patients already underwent surgery and thus were diagnosed T2 based on their surgical biopsies. This was the T2 EFRS group. A control group without surgery and thus without specific T1/T2 diagnosis was added.

We wanted to study if secretion analysis was able to show a T2 inflammatory pattern in a consistent and sensitive way, compared to the gold standard of surgical biopsies. We did not make a diagnosis of T1 or T2 CRS based on secretions, as we indeed observed control patients with a T2 inflammatory pattern on secretion analysis. However, we should be aware that the inflammatory pattern does not always reflect the symptoms present at that moment, and that a T2 pattern might also be present in e.g. allergic rhinitis. Moreover, the initiation of a T2 disease might go subclinical in a first phase as EFRS is usually diagnosed in an older population.

Peripheral blood counts were not included in the study as this was a clinical pilot study to compare different sampling methods. Consequently, ECP levels and other molecular analysis have not been determined as well. We focused on analysis of local pathology as we observed increasing evidence for cellular density in the target organs being more relevant than that in the blood compartment (e.g. Nair et al).

Question 3: Would it be possible for the authors to measure T1 and T2 cytokine levels in the nasal secretions? This in conjunction with the eosinophil counts would help to strengthen their claims of T2 or T1 status. If not this limitation should be discussed in the discussion. 

Answer: We agree with the reviewer that it seems a good idea to strengthen the T1 or T2 status by measuring specific cytokines levels. However, the current pilot study aimed at the evaluation of a low-cost test useful in daily clinical practice. The cost of the proposed test is <10€ in our experience and can be performed in all care centers providing pathology analysis. We have added a sentence about the possibility of local cytokine levels in the limitations section.

Question 4: For Acronyms that are repeated less than 10 times please consider writing them out in full. It is quite tiring for the reader to have to look up acronyms all the time. Or if preferred the authors could provide a list a used acronyms throughout the manuscript.

Answer: We have limited the number of abbreviations/acronyms and provided a list of used abbreviations at the end of the manuscript.

Question 5: Regarding Figures 1,2 and 3 from what source are these microscopy images from? Are they representative from ANS or NBS? Please include this in the figure legends

Answer: We have randomly selected the most visually attractive images. Both ANS and NBS are useful to detect the features of a T1 or T2 inflammatory pattern, as are their obtained images. However, ANS proved more sensitive in detecting T2 inflammatory signs and we included more ANS images than NBS images consequently. The origin of the images was added to the figure legends.

Question 6: Line 129: „No drug treatment changes were applied from secretions sampling to ESS.“ This sentence is not very clear. Do you mean no disease specific drug treatments were initiated between nasal secretion sampling and surgery?

Answer: Your interpretation is correct. No treatment alterations in the time frame between nasal secretion sampling and surgery were performed. We believe this is important for the readers when comparing nasal secretions and nasal biopsies. Especially no oral steroids were used. We have clarified this in the manuscript accordingly.

Question 7: Line 135: „The control patients were patients without chronic airway inflammatory disease clinically and endoscopically, but eventually a common cold” does not make sense and this is the wrong use of eventually. I presume the authors mean some of the control group had suffered symptoms of a common cold in the past year?

Answer: Thank you for notifying this. Controls had no chronic upper or lower airway inflammatory disease, not at the moment of inclusion and not by history taking. The only rhinological symptoms they had experienced in the past, were temporary symptoms matching a common cold, as you presumed correctly. We have modified this in the manuscript.

Question 8: Please define what is meant by the acronym EFRS.

Answer: EFRS stands for eosinophilic fungal rhinosinusitis. We have mentioned it in full upon first appearance and have added an abbreviation list.

Question 9: Throughout the article the authors repeat „on two different time points“ the correct preposition to use here is „at“ not „on“

Answer: Thank you, this has been modified throughout the manuscript.

Question 10: Line 206: please state in the text when these timepoints were. From table 3 I gather the samples were taken 3-4 days apart. Is this correct?

Answer: We have clarified this in the manuscript. Indeed, it concerns a measurement at day 0 and one at day +3 to +4.

Reviewer 3 Report

Ø  There are two abbreviations in the text used for the first time. They should be both written in full for the first time. The first one is “ESS” in the “Abstract” and the second one is “EFRS” in line 131.

Ø  In the “Abstract”, the aim of the study is included in the “Method”, while should be written in a different section.

Ø  Lines 90 and 104 need grammatical revision.

Ø  It should be made clear if the pathologist participating in this study was blinded to the patients’ clinical phenotypes or not.

Ø  There seems to be no information concerning the place of the study in the manuscript. It is best to be included in the methods section of both the “Abstract” and “Material and Methods”.

Author Response

Dear Reviewer

Thank you for your critical reading and suggestions to improve the paper. You may find the answers to your questions below.

Question 1: There are two abbreviations in the text used for the first time. They should be both written in full for the first time. The first one is “ESS” in the “Abstract” and the second one is “EFRS” in line 131.

Answer: EFRS stands for eosinophilic fungal rhinosinusitis. We have mentioned it in full upon first appearance and have added an abbreviation list. ESS stands for endoscopic sinus surgery, but we have decided to write this in full given its limited number of appearances.

Question 2: In the “Abstract”, the aim of the study is included in the “Method”, while should be written in a different section. 

Answer: We have moved the aim of the study from the Methods section to the end of the Introduction section.

Question 3: Lines 90 and 104 need grammatical revision.

Answer: Thank you, we have modified these sentences.

Question 4: It should be made clear if the pathologist participating in this study was blinded to the patients’ clinical phenotypes or not.

Answer: We admit it is interesting to add that the pathologist was indeed blinded for the phenotypes and have done this accordingly.

Question 5: There seems to be no information concerning the place of the study in the manuscript. It is best to be included in the methods section of both the “Abstract” and “Material and Methods”.

Answer: The study was performed in Belgium, Europe. All included patients were Caucasians. We have added this to the manuscript.

Reviewer 4 Report

This is a cross sectional study to assess the usefulness of nasal blown secretion and aspiration of nasal secretion to assess for type 1 or type 2 inflammation among chronic rhinusinunsitis patients.

Introduction:

Please introduce and spell out EFRS at first mention in introduction to initiate readers.

Methods:

Line 136: Please clarify if in second experiment the contro population had common cold during sample collection

Line 141: Please add the local ethics board approval number

Line 158: please add the handling of nasal secretion samples prior to being embedded in paraffin. Please explain if there were any thin secretions which diluted in the thin prep and if so how this was dealt with.

Results

Line 174: Please provide the tissue definition of T1 and T2 inflammation

Discussion:

Please add explanation why EFRS patients 1 year post surgery was selected to assess for reproducibility of NBS and ANS and discuss the limitations of this method ie. No tissue biopsy at time of secretion sampling.

Author Response

Dear Reviewer

Thank you for your critical reading and suggestions to improve the paper. You may find the answers to your questions below.

Question 1: Introduction: Please introduce and spell out EFRS at first mention in introduction to initiate readers.

Answer: EFRS stands for eosinophilic fungal rhinosinusitis. We have mentioned it in full upon first appearance and have added an abbreviation list.

Question 2: Methods: Line 136: Please clarify if in second experiment the control population had common cold during sample collection.

Answer: Controls had no chronic upper or lower airway inflammatory disease, not at the moment of inclusion and not by history taking. The only rhinological symptoms they had experienced in the past, were temporary symptoms matching a common cold. We have modified this in the manuscript.

Question 3: Line 141: Please add the local ethics board approval number

Answer: We have added the local ethics board approval number.

Question 4: Line 158: please add the handling of nasal secretion samples prior to being embedded in paraffin. Please explain if there were any thin secretions which diluted in the thin prep and if so how this was dealt with.

Answer: Upon collection, Cytolyt solution was added to the nasal secretions. In this way, the morphology of general cytology cellular samples is preserved and fixated, and mucus is dissolved. Samples underwent routine workflow for paraffin embedding. Samples with thick mucus and thin secretions were treated the same way in order not to interfere with cellular concentration. As stated above, Cytolyt contains a mucolytic which dissolves thick mucus without interfering with cellular content. In our experience, patients with thin or few secretions usually did not exhibit a severe phenotype based on endoscopy, which was confirmed by blinded structured histopathology.

Question 5: Results: Line 174: Please provide the tissue definition of T1 and T2 inflammation

Answer: According to the EPOS 2020 guidelines a tissue T2 endotype was defined by the presence of ten or more eosinophils per high power field (400x magnification) in a Caucasian population. A T1 inflammatory pattern was retained on the finding of neutrophilic cells in combination with an eosinophilic cell count less than 10 per high power field. This has been clarified in de manuscript.

Question 6: Discussion: Please add explanation why EFRS patients 1 year post surgery was selected to assess for reproducibility of NBS and ANS and discuss the limitations of this method ie. no tissue biopsy at time of secretion sampling.

Answer: All EFRS patients had a proven T2 inflammatory pattern based on pathology results of tissue analysis. By implementing a postoperative interval of more than one year, we wanted to avoid eventual effects of direct surgical influence or post-surgical healing process. Moreover, the EFRS subgroup is known for a high recurrence risk and needs a lifetime medical follow-up. Hence, we tried to find parameters which could predict or demonstrate ongoing or imminent activation of disease, even when no recurrence could be seen on endoscopic examination. As literature does not include studies with postoperative follow-up by secretion samplings or tissue biopsies, we aimed to demonstrate the potential role of nasal secretion sampling and subsequent structured pathology analysis. We are aware that the results of the second experiment could not be compared with tissue analysis, but the latter is more invasive. Moreover, in the first experiment with a much larger sample size than the second, we were able to demonstrate that nasal secretion samplings, especially obtained via the nasal aspiration technique, is a sensitive and very specific test compared to nasal biopsy analysis. Consequently, this is the reason why we did not perform tissue biopsies in the second experiment, including patient who did not need surgery. However, we understand your concern and added this to the limitations section. Eventually, we hope that nasal secretion analysis can become a valid alternative for more invasive sampling methods at a moment when patients do not need surgery, especially at baseline given the therapeutical consequences, and aimed to stimulate further research about the topic with this pilot study.

Reviewer 5 Report

General comments

This manuscript is an article about usefulness of nasal secretions sampling

for endotyping chronic rhinosinusitis.

This is a novel and very fascinating article .

However, there is an improvement that should be made. 

Specific comments

In Table2, authors categorized eosinophil count by <10 eos/HPF, 10-49 eos/HPF, and

>99 eos/HPF.

In the JESREC study (Reference 3), Tokunaga et al categorized eosinophil count by <70 eos/HPF and >70 eos/HPF.

Please let me have the author’s comment.

I hope that my comment is useful for the improvement of this manuscript.

Author Response

Dear Reviewer

Thank you for your critical reading and suggestions to improve the paper. You may find the answers to your question below.

Question 1: In Table 2, authors categorized eosinophil count by <10 eos/HPF, 10-49 eos/HPF, and >99 eos/HPF. In the JESREC study (Reference 3), Tokunaga et al categorized eosinophil count by <70 eos/HPF and >70 eos/HPF. Please let me have the author’s comment.

Answer: Thank you for this interesting point. The authors of the JESREC study implemented a diagnostic concept aiming at delineating a severe T2 inflammatory endotype, with a correlation to prognosis in the Japanese population. Our pilot study follows the definition proposed by the EPOS 2020 guidelines to accept the diagnosis of a T2 endotype when eosinophil numbers are equal or more than 10 eosinophils per high power field (400x magnification). Both criteria have a purpose: the JESREC study aimed to delineate the more aggressive T2 endotypes whereas the EPOS study aimed to establish a T2 diagnostic setting in the Caucasian race. Our study rather leans towards the latter, not only regarding the race but also regarding the intention: delineating a group with an underlying pathology typical for a T2 inflammatory pattern. This is the reason why we selected the EPOS cut-off. However, in our experience, we also notice a correlation between a higher eosinophil count and worse prognosis, which was not the aim of the current pilot study though.